# Phenotypes of Chronic Rhinosinusitis and Peripheral Blood Leukocytes Parameters in Elderly Patients

**DOI:** 10.3390/medicina59010126

**Published:** 2023-01-09

**Authors:** Grażyna Stryjewska-Makuch, Joanna Glück, Olga Branicka, Grażyna Lisowska

**Affiliations:** 1Department of Laryngology and Laryngological Oncology, Leszek Giec Upper-Silesian Medical Centre of the Silesian Medical University in Katowice, 40-635 Katowice, Poland; 2Department of Internal Diseases, Allergology and Clinical Immunology, Medical University of Silesia, 40-752 Katowice, Poland; 3Department of Otorhinolaryngology and Laryngological Oncology in Zabrze, Medical University of Silesia, 40-635 Katowice, Poland

**Keywords:** chronic rhinosinusitis, patients over 65 years of age, neutrophil-to-lymphocyte ratio

## Abstract

*Background and Objectives*: Chronic rhinosinusitis (CRS) is a common disease that can be differentiated into two phenotypes, with or without polyps (CRSwNP) or CRSsNP), which may be unilateral (UNIL) or bilateral (BIL). CRS may have an impact on absolute neutrophils and lymphocytes count in peripheral blood. The aim of the study was to investigate whether the incidence of a specific CRS phenotype changes with age and to compare the values of neutrophils, lymphocytes and neutrophil-to-lymphocyte ratio in the peripheral blood between groups of patients below and above 65 years of age with different CRS phenotypes. *Material and Methods*: A total of 235 patients aged 65 and over were examined, including 140 (59.6%) males. The group of patients <65 years of age comprised 160 subjects, including 103 (64.4%) males. In both groups, the sequence of frequency of particular phenotypes was similar: the most common phenotype was bilateral CRSwNP followed by CRSsNP BIL, CRSsNP UNIL, and finally, CRSwNP UNIL. Direct comparisons between determined phenotype in both groups of different ages revealed that, in the group ≥65 years, CRSwNP BIL occurred significantly more often than in the group <65 years of age. In fact, in the <65 group, bilateral CRSsNP was more common. The absolute neutrophils and lymphocytes counts were significantly higher in the whole group of patients with CRS ≥65 years of age and absolute number of neutrophils was higher in ≥65 years of age group with bilateral CRSsNP. *Conclusions*: The higher number of neutrophils in the whole ≥65 years of age group and in older patients with bilateral CRSsNP may indicate that CRS, despite of phenotype, may be an important source of infection that requires surgical treatment in elderly patients as well.

## 1. Introduction

Chronic rhinosinusitis (CRS) is a heterogeneous inflammatory disease of the airways with complex pathophysiology. CRS, diagnosed according to the criteria of the European Position Paper on Rhinosinusitis and Nasal Polyps 2020 (EPOS 2020) [1], affects 10.9% of the population in Europe [2] and 12% in the United States [3]. CRS occurs at any age, regardless of phenotype. It was observed in the U.S. that the CRS phenotype with polyps (CRSwNP) was significantly more common in patients over 40 years of age [4]. Research conducted in Korea showed that the incidence of CRSwNP increased with age and was almost twice as high among patients over 60 years of age than in those under 40 years of age [5]. The recently reported prevalence of CRS among people >60 years old was 4.7%, with CRS emerging as the sixth most common chronic condition in the elderly [6,7]. Age-related impairment of innate defence mechanisms (immunosenescence) reduces phagocytosis of the pathogenic bacterial flora, and also lowers the level of the S100A8/S100A9 protein, which is essential for the proper functioning of the epithelial barrier [8]. The reduction in the number and frequency of bending of the cilia on the sinus mucosa results in impaired mucociliary clearance. An increase in the level of pro-inflammatory cytokines, i.e., interleukin (IL)-beta, IL-6, chemokine IL-8 and tumour necrosis factor alpha (TNF-alpha), has been shown in patients over 60 years of age affected by CRS. This causes tissue neutrophilia and excessive bacterial colonization of the sinus mucosa, recurrent infections with poor long-term response to antibiotics and increased steroid resistance [9]. Other systemic parameters, such as CRP and leukocytosis, in patients qualified for endoscopic surgery should, by definition, be within normal limits to avoid postoperative complications. Peripheral eosinophilia seems to be a parameter independent of the age of patients only for the CRS endotype. The neutrophils and lymphocytes counts, as well as the neutrophil-to-lymphocyte ratio (NLR) in the peripheral blood, are among the parameters that may vary depending on the extent of the inflammation [10,11,12]; this is also true in the paranasal sinuses.

The aim of the study was to investigate whether the CRS phenotype changes with age. The results verified which of the primary phenotypes of chronic rhinosinusitis, unilateral or bilateral, was the most common in the group of patients over 65 years of age compared to the group of younger patients. Moreover, considering that NLR may reflect inflammatory processes that are ongoing in different places, including in the paranasal sinuses, the values of neutrophils, lymphocytes and peripheral blood NLR between CRS phenotypes were assessed and compared in the above-mentioned groups of patients.

## 2. Materials and Methods

We conducted a retrospective single-centre study. The study group consisted of all patients aged over 65 years who met the inclusion criteria and were operated on for chronic rhinosinusitis (CRS) in the years 2016–2020 at the Department of Laryngology and Laryngological Oncology of the Upper Silesian Medical Centre in Katowice Ochojec. The patients were qualified for the procedure in accordance with the EPOS 2020 guidelines. The control group consisted of patients under 65 years of age who were treated for CRS in 2020 (from 1 January to 31 December). All patients had previously been treated on an outpatient basis for at least 12 weeks with maximal pharmacological therapy, and in the absence of improvement, they were qualified for sinus surgery.

All patients were assessed for demographic data, comorbidities such as asthma, IgE-dependent allergy, hypersensitivity to nonsteroidal anti-inflammatory drugs (NSAIDs), diabetes mellitus, hypertension, and smoking. Independently of CRS endotype, patients were divided into groups with or without polyps. The following CRS phenotypes were recognised: phenotype of CRS with unilateral or bilateral polyps (CRSwNP UNIL or BIL) and phenotype with unilateral or bilateral localisation of inflammatory lesions, but without polyps (CRSsNP UNIL or BIL). Endoscopic examination of the nasal cavity was also performed. In the case of CRSwNP, polyps were assessed according to the Lund-Kennedy scoring system.

Regardless of whether an eosinophilic endotype, eosinophilic tissue infiltrates or peripheral eosinophilia associated with N-ERD, atopy or asthma were present, we focused only on neutrophil and lymphocyte counts as a systemic indicator of inflammation. The lymphocytes and neutrophils counts, as well as the NLR value, were obtained from the tests performed during the first day of hospitalization.

Blood samples were collected into a hematologic sample tube containing anticoagulant, and neutrophil and lymphocyte absolute counts were recorded using an Haematology Analyzer Sysmex XN-1000 (Sysmex Europe Corporation, Norderstedt, Germany) haematology analyser. Using these data, the neutrophil absolute number was divided by the lymphocytes absolute number, and NLR ($\frac{neutrophil\;absolute\;number}{lymphocyte\;absolute\;number})$ values were calculated.

### 2.1. Inclusion Criteria

Patients reporting nasal congestion and/or nasal disturbances combined with facial pain and/or olfactory disorder lasting over 12 weeks, despite applying the maximal pharmacological treatment according to the EPOS 2020 guidelines, were qualified for ESS. Patients did not report any exacerbation of symptoms and did not undergo antibiotic therapy or systemic steroid therapy for at least 4 weeks before the surgery. All patients used topical nasal steroids up to operation point. The diagnosis of IgE-dependent allergy was based on the history confirmed by the prick test and/or the result of testing for the presence of specific IgE antibodies. Asthma was diagnosed according to the guidelines of the Global Initiative for Asthma (GINA) [13]. Patients used inhalation therapy, and on admission to the department asthma was well-controlled. The authors have used the term ‘hypersensitivity to NSAIDs’, which is defined by the European Academy of Allergology and Clinical Immunology (EAACI) as objectively reproducible symptoms that are caused by exposure to a specific stimulus at a dose tolerated by other people [14]. No provocation tests with NSAIDs were performed, and qualification was based on the anamnestic data. In patients qualified for surgery under general anaesthesia, comorbidities other than asthma (HA, DM) were required to be well controlled. Patients with allergy were not operated on during a pollen season. The extent of the inflammation in the paranasal sinuses was determined on the basis of computed tomography using the Lund-Mackay staging system.

### 2.2. Exclusion Criteria

Patients under 18 years of age, pregnant women, patients with secondary CRSwNP associated with systemic diseases (e.g., eosinophilic granulomatous vasculitis, granulomatous vasculitis, fungal sinusitis, sarcoidosis, primary ciliary dyskinesia, and cystic fibrosis), as well as patients who reported exacerbation of symptoms, or who took antibiotics or systemic steroids less than 4 weeks before the surgery, were excluded from the study.

The study protocol was reviewed and approved by the Bioethics Committee at the Silesian Medical University in Katowice (PCN/CBN/0052/KB/165/22), complied the ethical rules for human experimentation that started in the Declaration of Helsinki, and informed consent for this analysis was waived because of the retrospective nature of the study. In the analysis, confidentiality of patient’s health information was maintained, and analysis was performed using anonymized dataset.

### 2.3. Statistical Analysis

Results are expressed as absolute numbers and percentages for frequencies, and median values with interquartile and total ranges for the other parameters. The nonparametric Mann–Whitney U rank sum test for comparisons between non-related groups and Z-test for proportions for comparisons between frequencies were used. All analyses were performed with a software package (The STATISTICA 13.3, StatSoft, Kraków, Poland). *p* values less than 0.05 were considered significant.

## 3. Results

The group of Caucasian patients over 65 years of age consisted of 235 individuals, including 140 (59.6%) males and 95 (40.4%) females; the median age was 68 years (range 65–84, interquartile range 66–71). The group of patients under 65 years of age consisted of 160 (64.4%) individuals, 103 males and 57 (35.6%) females; the median age was 44 years (range 23–64, interquartile range 37–61). In both groups, the frequency of phenotypes was similar: the most common phenotype was CRSwNP BIL, followed by CRSsNP BIL, CRSsNP UNIL, and finally, CRSwNP UNIL.

In the <65 group, CRSsNP BIL occurred significantly more often than in the ≥65 group, and in the ≥65 group, CRSwNP BIL occurred significantly more often than in the <65 group. The frequency of unilateral lesions was comparable in both age groups (Figure 1).

The neutrophils and lymphocytes counts, as well as NLR, were compared between the whole groups as well as phenotypes in the groups ≥ and <65 years of age (Table 1). The neutrophils and lymphocytes counts were significantly higher in the group ≥65 years of age (Figure 2 and Figure 3).

When comparing the distinct phenotypes, only the group of patients with the CRSsNP BIL phenotype had a neutrophil count that was significantly higher in the group of patients ≥65 than in young group.

The neutrophils and lymphocytes counts, as well as NLR, were compared within the groups ≥65 and <65 years of age. The examined parameters did not differ significantly in the studied subgroups within the ≥65 group. Within the <65 group, the number of neutrophils was higher in CRS wNP BIL than in CRSs NP UNIL (*p* = 0.03) and CRSs NP BIL (*p* = 0.047) (Figure 4).

Among comorbidities, allergy was more common in the <65 group, and asthma and smoking were comparable in both groups. NSAIDs hypersensitivity, arterial hypertension, diabetes mellitus and neoplasm were more common in the ≥65 group (Table 2).

In the group of patients younger than 65 years, 36 patients (22%), and in the older patients’ group, 101 patients (43%) had received previous operations on at least one occasion (*p* < 0.0001).

## 4. Discussion

The aim of the study was to investigate whether the CRS phenotype changes with age in patients qualified for endoscopic sinus surgery. The phenotypes of chronic rhinosinusitis, such as with or without polyps and unilateral or bilateral, were measured in the group of patients over 65 years of age and compared to the group of younger patients. Moreover, absolute counts of peripheral blood neutrophils and lymphocytes were assessed in patients with different CRS phenotypes and between the two groups.

The present study found that the CRSwNP phenotype was the most common irrespective of age, which is consistent with the available literature [4,5]. Direct comparisons of the percentage of patients with CRSwNP between both groups revealed that this phenotype was more common within the group ≥65 age of age than in the younger group (64.7% vs. 51.3% *p* = 0.008).

In the studied group of patients under 65 years of age, the CRSsNP phenotype was more common than in the older group. Such observations may suggest that CRSwNP in some patients may develop with age and that increasing inflammation causes a smooth transition from one phenotype to another.

In our study we found that the level of peripheral blood neutrophils and lymphocytes was the significantly higher in the whole group of patients with CRS and ≥65 age than in the younger group. It is worth underlining that eosinophilic disorders, such as NSAIDs hypersensitivity [14], were more common in patients in the group ≥65 of age, but inhalant allergies were more common in the younger group; asthma was equally common in both groups. Thus, our finding of a higher absolute neutrophil count in older patients with CRSsNP phenotype is interesting and unexpected.

CRSwNP is a phenotype that has the greatest extent of inflammatory lesions, is difficult to treat, and has frequent exacerbations and recurrence of symptoms. Reports in the literature [7,15] suggest that in patients over 65 years of age, there are frequent neutrophilic infiltrates within polyps, and in the case of polyps with tissue eosinophilia, a decreased level of eosinophil cationic protein (ECP) is observed, correlating with the age of patients [14]. This may be the cause of the difficulties encountered in the conservative treatment of CRSwNP in elderly patients. Moreover, in patients over 65 years of age, there was a decreased level of S100A8/A9 (calprotectin) [16], which resulted in the weakening of the epithelial barrier and activity against Staphylococcus aureus, Klebsiella species and fungi [16]. All the above-mentioned observations suggest the existence of a significant focus of infection within the sinuses, which, especially in patients ≥65 years of age, may favour the occurrence of other systemic diseases [17,18,19,20,21].

Elderly patients with CRS often complain of non-specific symptoms such as cough, anosmia, rhinorrhoea, and a feeling of pressure or pain in the face, as well as concentration disorders or slow thinking and slow response [22], which in combination with often numerous comorbidities, may be the cause of late qualification for surgical treatment. It has not been found that postoperative complications occur more frequently with the age of patients, apart from the more frequent use of anticoagulants in the group of older patients, which requires more careful preparation for surgery [23,24]. Similarly, Vleming et al. [25] has shown that complications after endoscopic surgery are independent of the patient’s age. It is important to remember that endoscopic sinus surgery requires general anaesthesia with controlled negative pressure, which may pose additional cardiovascular risks in elderly patients irrespective of the burden of other comorbidities, which holds some physicians and patients back from the decision to undertake surgery. A high neutrophils count may provide an additional argument for surgery. The argument in favour of the surgical treatment is the lower frequency of recurrences after endoscopic sinus surgery for CRSwNP in elderly patients, even in the eosinophilic endotype [25]. Ban et al. have observed that diabetes mellitus and a longer duration of symptoms before sinus surgery correlate positively with the frequency of complications [26], which may be an argument for earlier surgical intervention. Research conducted by Yancey [27] has shown a significant impact of CRS on the quality of life of elderly patients and, unfortunately, a lower impact of surgery on improving the quality of life compared to the younger group. The surgery reduces the risk of local and systemic complications (cardio-vascular, stroke) especially in patients with a higher blood neutrophil absolute count.

A limitation of the study is that it was not possible to investigate the absolute neutrophils and lymphocytes counts or the NLR score value after sinus surgery. Unilateral lesions were observed equally often in both age groups, but the groups of patients with unilateral phenotypes were small, so our conclusions must be only initial and applied cautiously. In addition, it would have been useful to track whether the peripheral blood parameters in individual CRS phenotypes are related to the L–M score value. These problems require further research.

## 5. Conclusions

In patients ≥65 years of age, the phenotype of CRS with polyps was the most common. The neutrophil count was higher in the bilateral phenotype without polyps in elderly patients. The obtained results suggest that, regardless of phenotype, CRS may be a significant infectious focus that requires surgical treatment in elderly patients as well younger ones, after careful preparation for general anaesthesia.

## Figures and Tables

**Figure 1 medicina-59-00126-f001:**
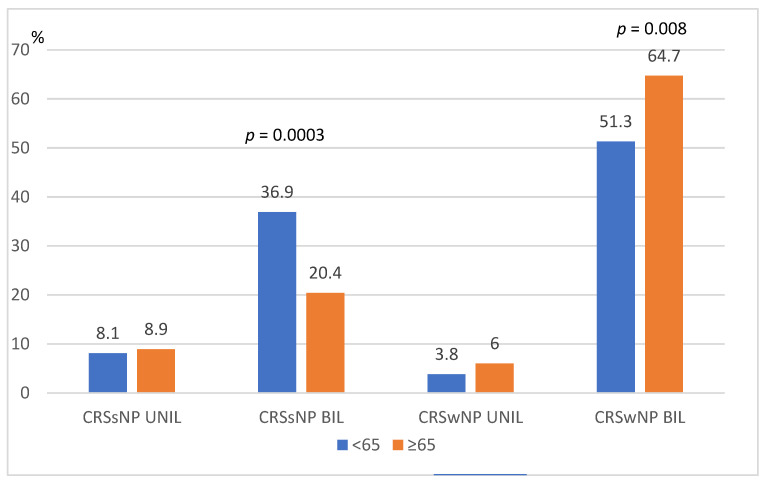
Rhinosinusitis phenotypes in groups of patients < and ≥65 years of age.

**Figure 2 medicina-59-00126-f002:**
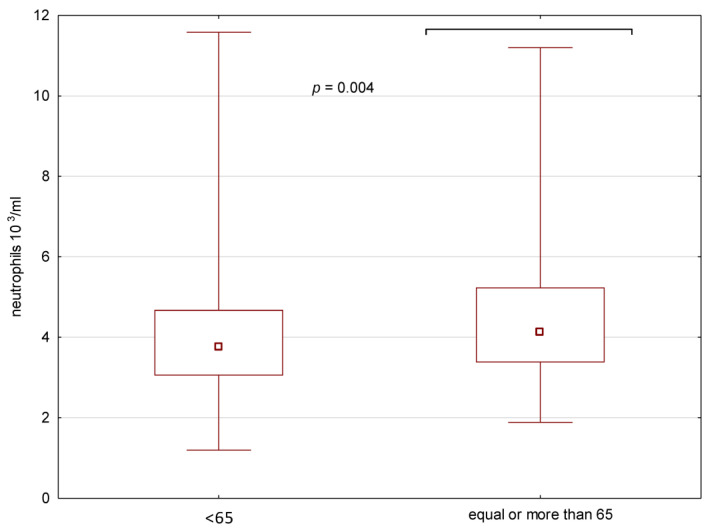
Neutrophils in the peripheral blood in the two groups of patients (≥65 and <65 years of age).

**Figure 3 medicina-59-00126-f003:**
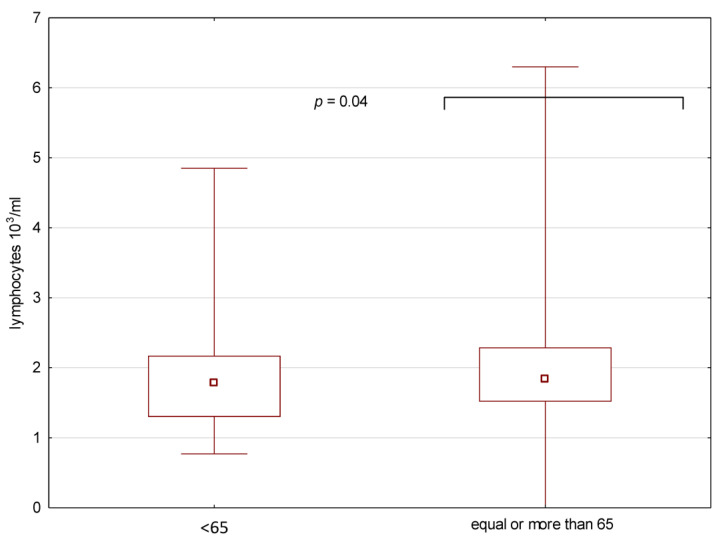
Lymphocytes in the peripheral blood in the two groups of patients (≥65 and <65 years of age).

**Figure 4 medicina-59-00126-f004:**
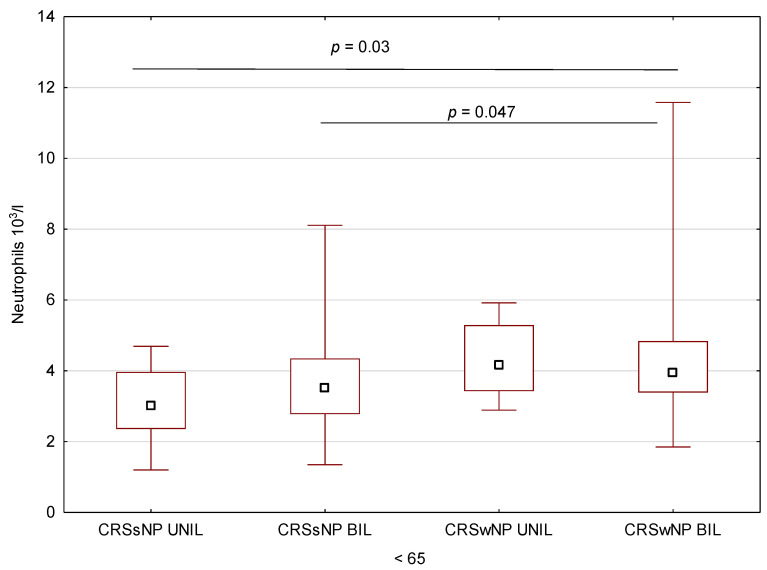
Neutrophils count in the group of patients <65 years of age for individual phenotypes.

**Table 1 medicina-59-00126-t001:** Assessment median and IQR of neutrophils, lymphocytes and NRL in various phenotypes in the groups ≥ and <65 years of age.

	<65 Years	≥65 Years	*p* *
Total	n = 156	n = 217	
Neutrophils, 10^3^/µL	3.76 (3.05–4.68)	4.14 (3.35–5.13)	0.004
Lymphocytes, 10^3^/µL	1.83 (1.30–2.18)	1.84 (1.53–2.29)	0.04
NLR	2.15 (1.64–2.88)	2.24 (1.65–2.89)	NS
Phenotype CRSsNP UNIL	n = 10	n = 20	
Neutrophils, 10^3^/µL	3.00 (2.36–3.97)	3.95 (3.26–4.58)	NS
Lymphocytes, 10^3^/µL	1.89 (1.24–2.06)	2.03 (1.61–2.38)	NS
NLR	1.87 (1.19–2.37)	2.03 (1.49–2.63)	NS
Phenotype CRSsNP BIL	n = 59	n = 43	
Neutrophils, 10^3^/µL	3.52 (2.78–4.35)	4.07 (3.27–4.94)	0.034
Lymphocytes, 10^3^/µL	1.76 (1.32–2.16)	1.79 (1.60–2.35)	NS
NLR	1.99 (1.59–2.60)	2.10 (1.64–2.90)	NS
Phenotype CRSwNP UNIL	n = 6	n = 13	
Neutrophils, 10^3^/µL	4.15 (3.43–5.29)	4.57 (2.91–5.28)	NS
Lymphocytes, 10^3^/µL	1.59 (1.12–1.98)	1.72 (1.49–2.05)	NS
NLR	2.88 (1.74–4.80)	2.57 (1.55–3.72)	NS
Phenotype CRSwNP BIL	n = 78	n = 136	
Neutrophils, 10^3^/µL	3.95 (3.39–4.84)	4.10 (3.37–5.07)	NS
Lymphocytes, 10^3^/µL	1.86 (1.31–2.26)	1.84 (1.47–2.23)	NS
NLR	2.34 (1.67–3.02)	2.29 (1.68–2.96)	NS

CRSsNP—chronic rhinosinusitis without nasal polyps; CRSwNP—chronic rhinosinusitis with nasal polyps; BIL—bilateral; UNIL—unilateral; NLR—neutrophil-to-lymphocyte ratio; NS—non-significant. * U Mann–Whitney test; data are shown as median and interquartile range.

**Table 2 medicina-59-00126-t002:** Frequencies of comorbidities in both groups of patients.

	<65, n = 160	≥65, n = 235	*p*
NSAIDs hypersensitivity	14 (26.3%)	51 (73.4%)	0.0008
asthma	43 (26.9%)	65 (27.7%)	ns
inhalant allergy	71 (44.4%)	38 (16.2%)	0.00001
arterial hypertension	42 (26.3%)	173 (73.6%)	0.00001
diabetes mellitus	17 (10.6%)	52 (22.1%)	0.0035
smoking	26 (10.6%)	35 (14.9%)	ns
neoplasm	2 (1.3%)	18 (7.7%)	0.005

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
