# Peer review of "Phenotypes of Chronic Rhinosinusitis and Peripheral Blood Leukocytes Parameters in Elderly Patients"

_medicina, 2023, doi:10.3390/medicina59010126_

Round 1
Reviewer 1 Report (Previous Reviewer 1)
Dear Author,
This article entitled Phenotypes of chronic rhinosinusitis and peripheral blood leukocytes parameters in elderly patients is interesting. I have read the paper and found it as a potential article for consideration for publication. However, there are some specific points that should be corrected before taken such decision.
1. The detail sample size calculation methodology should be reported.
2. This value p=0.0003 should be reported as p<0.001.
3. The authors should inform the average time of disease, and use of medication.
4. What are the other confounding factors that might be involved in the pathophysiology of chronic rhinosinusitis in elderly patients???
5. I suggest authors to test multivariate model to see which variables are exerting more influence on symptoms. Also, can authors perform a ROC curve?
Author Response
Review 1
- The detail sample size calculation methodology should be reported.
- This value p=0.0003 should be reported as p<0.001.
- The authors should inform the average time of disease, and use of medication.
The average time of disease, we do not have this data All patients had previously been treated on an outpatient basis for at least 12 weeks with maximal pharmacological therapy wich included treatment with intranasal steroids a minimum 12 weeks, short courses of antibiotic therapy ( from 1-5 courses) 1 to 4 short inserts of systemic steroid.
- What are the other confounding factors that might be involved in the pathophysiology of chronic rhinosinusitis in elderly patients???
Factors that may have been involved in the pathophysiology of chronic rhinosinusitis may have been working conditions, the environment in which patients spent the most time, material status, diet, medications and stimulants used.
- I suggest authors to test multivariate model to see which variables are exerting more influence on symptoms. Also, can authors perform a ROC curve?
The aim of this study was to compare a prevalence of different CRS phenotypes in groups of patients stratified based on age. We had not assessed symptoms. However , according to the reviewer’s suggestion we performed multivariate test and assessed the influence of following variables: sex, diabetes, allergy, asthma, arterial hypertension, NSAIDs hypersensitivity. We found that in the group of patients younger than 65 years with CRSsNP BIL phenotype neutrophils number was related to occurrence of arterial hypertension (R = 0.09; F = 2.588, p = 0.023). Because it was the only significant relation and it was not repeated in any other groups we did not show this results in the manuscript.

Reviewer 2 Report (New Reviewer)
Dear Authors.
This is an interesting study and I find the study design acceptable.
However, there are a few things that need clarification.
Major comments
1. With regard to the methods, what are the clear criteria for determining whether polyps and inflammatory lesions are unilateral or bilateral in the CRS phenotype, and did you also refer to the Lund-Mackay system? Was there also a decision made by the otolaryngologist?
2. This study is supposed to include eosinophilic sinusitis in CRS. Eosinophilic sinusitis and neutrophilic sinusitis have different characteristics. Eosinophilic sinusitis is considered to be cribriform sinus predominant, but have you had any cases where it was difficult to determine whether it was bilateral or unilateral?
3. As a complication, was there any diffuse panbronchiolitis characteristic of sinusitis? If yes, please add to table 3.
4. The exclusion of fungal sinusitis may also need to be mentioned in the text.
5. With regard to NLR, did a history of smoking affect neutrophil and lymphocyte counts, which was not an issue in the analysis of this study? You state that smoking was comparable in the groups aged 65+ and <65 years, please explain in more detail.
Author Response
Review 2
- With regard to the methods, what are the clear criteria for determining whether polyps and inflammatory lesions are unilateral or bilateral in the CRS phenotype, and did you also refer to the Lund-Mackay system? Was there also a decision made by the otolaryngologist?
The decision was made by an ENT specialist (first author).
The sinuses were assessed in three planes. The inflammatory lesions were assessed according to the Lund-Mackay score (the degree of opacification in the maxillary, anterior and posterior ethmoid, frontal, and sphenoid sinuses as well as the obstruction of the ostiomeatal complexes were evaluated (on both sides) on a 0-2 scale - a maximum of 24 points). Unilateral inflammatory lesions occurred only in the sinuses on the right or left side, bilateral lesions occurred in varying degrees in the paranasal sinuses always on both sides. CRS was qualified for the phenotype with polyps when polyps were visible on endoscopic examination bilaterally, at least in the middle nasal meatus (Lund-Kennedy score).
- This study is supposed to include eosinophilic sinusitis in CRS. Eosinophilic sinusitis and neutrophilic sinusitis have different characteristics. Eosinophilic sinusitis is considered to be cribriform sinus predominant, but have you had any cases where it was difficult to determine whether it was bilateral or unilateral?
If the inflammatory lesions involved the ethmoid cells bilaterally (independently of the other sinuses) this type of inflammation was referred to as bilateral lesions.
- As a complication, was there any diffuse panbronchiolitis characteristic of sinusitis? If yes, please add to table 3.
Patients qualified for endoscopic surgery did not have panbronchiolitis.
- The exclusion of fungal sinusitis may also need to be mentioned in the text.
We agree with the Reviewer
- With regard to NLR, did a history of smoking affect neutrophil and lymphocyte counts, which was not an issue in the analysis of this study? You state that smoking was comparable in the groups aged 65+ and <65 years, please explain in more detail.
The numbers of neutrophils and lymphocytes as well as NLR were compared within the groups ≥65 and <65 years of age.
Smoking were comparable in both groups, in group <65 smoked 10,6%, in group>65 14,9% patients hence smoking was considered to have no significant effect on the results. . Exact data are included in the tab. 3 – previously, now table 2.
Reviewer 3 Report (New Reviewer)
The authors have presented a study about "Phenotypes of chronic rhinosinusitis and peripheral blood leukocytes parameters in elderly patients". The manuscript is interesting, investigating CRS phenotypes and blood leukocytes according to the age group. There are some suggestions that could be made to increase the overall quality.
1. The authors focused only on neutrophil and lymphocyte count without eosinophil count as a systemic indicator of inflammation. However, serum eosinophil count (tissue eosinophil count, if possible) or serum total IgE must be included because eosinophilic CRS and Th2 inflammation are dominant in the western population, who were participants in the present study. If evaluating eosinophil count is impossible, the terminology of CRS should be changed to another term such as “non-eosinophilic CRS”.
2. Although the authors explained systemic parameters in the introduction section, there is still a minor lack of information regarding the role of laboratory values, particularly in CRS.
3. Other laboratory parameters of inflammation such as WBC, ESR, CRP, procalcitonin, LUC (large unstained cells), and DNI (delta neutrophil index) in CRS should be discussed in detail in the discussion section.
4. Systemic parameters could not reflect the local inflammatory status of the sinonasal area. This aspect should be added in the discussion section.
5. As the authors explained, CRS is a multifactorial disease. Therefore, there is a possibility of racial or endotypic differences. However, the data included in this study might be skewed in terms of race and endotype. In addition, there was no information regarding primary or recurrent cases. Please describe these aspects in discussion section.
6. Higher neutropihls in older patients with CRSsNP was an interesting and unexpected finding, as the authors mentioned. Please explain the reason and provide an interpretation of this finding thoroughly in the discussion section.
7. Please unify the acronyms of Unilateral (UNIL) or bilateral (BIL). For example, UNI and BIL.
8. It seems that Table 1 is redundant because it is exactly the same as Figure 1. Please consider removing one of them. (probably table 1)
9. Figure 1: Please provide the unit of the Y-axis. (probably percentage)
10. Figure 2 and 3: Please revise the order and the name in the X-axis. (<65 years first and ≥ 65 years later, in addition please note “≥”, not “>”)
11. It is hard to find the description in the text regarding Figure 3. Please add it in detail.
12. Figure 4: It looks like there are statistical differences between CRSsNP UNIL and CRSwNP UNIL in addition to CRSsNP UNIL and CRSsNP BIL. Please clarify this.
13. The authors set the age of 65 as the cut-off line. Please add the reference from the previous literature explaining the cut-off of old age.

Author Response
Reviewer 3
- The authors focused only on neutrophil and lymphocyte count without eosinophil count as a systemic indicator of inflammation. However, serum eosinophil count (tissue eosinophil count, if possible) or serum total IgE must be included because eosinophilic CRS and Th2 inflammation are dominant in the western population, who were participants in the present study. If evaluating eosinophil count is impossible, the terminology of CRS should be changed to another term such as “non-eosinophilic CRS”
Of course, we agree that in the European, Caucasian population, a Th2-dependent eosinophilic profile predominates in CRSwNP. Each of the operated patients had polyps collected for histopathological examination, and by far the most common was eosinophilic infiltration in the examined tissues with peripheral eosinophilia. Hence, we cannot use the term "non-eosinophilic CRS." We were all the more interested in the image of the remaining cells especially the neutrophil and lymphocyte count.
- Although the authors explained systemic parameters in the introduction section, there is still a minor lack of information regarding the role of laboratory values, particularly in CRS.
In CRS, we usually test eosinophil levels, especially when we consider qualifying patients for steroid therapy or biologic therapy. Assessment of serum IgE is important in atopic patients in whom we are considering allergen immunotherapy.
- Other laboratory parameters of inflammation such as WBC, ESR, CRP, procalcitonin, LUC (large unstained cells), and DNI (delta neutrophil index) in CRS should be discussed in detail in the discussion section.
All operated patients were prepared for surgery using maximal medical treatment , in laboratory tests the values of WBC, ESR ((the erythrocyte sedimentation rate - ESR)), CRP were normal. Other parameters (DNI, LUC) were not evaluated.
- Systemic parameters could not reflect the local inflammatory status of the sinonasal area. This aspect should be added in the discussion section.
Since there are reports of the impact of chronic inflammatory conditions on the whole body, it has been suggested that elevated NLR values may reflect the impact of CRS on overall health, among other things.
- Stryjewska-Makuch G, Glück J, Niemiec-UrbaÅ„czyk M, et al. Inflammatory lesions in the paranasal sinuses in patients with ischemic stroke who underwent mechanical thrombectomy. Pol Arch Intern Med. 2021;131:326-331.
- Agca MC, Aksoy E, Duman D, et al. Does eosinophilia and neutrophil to lymphocyte ratio affect hospital re-admission in cases of COPD exacerbation? Tuberk Toraks. 2017;65(4): 282-290.
- Hendy RM, Elawady MA, Mansour AI. Assessment of neutrophil /lymphocyte percentage in bronchial asthma. Egypt J Chest Dis Tuberc. 2019;68:74-79.
- Mochimaru T, Ueda S, Suzuki Y et al. Neutrophil-to-lymphocyte ratio is a novel independent predictor of severe exacerbation in asthma patients. Ann Allergy Asthma Immunol. 2019;122(3):337-339.
- As the authors explained, CRS is a multifactorial disease. Therefore, there is a possibility of racial or endotypic differences. However, the data included in this study might be skewed in terms of race and endotype. In addition, there was no information regarding primary or recurrent cases. Please describe these aspects in discussion section.
All subjects were Caucasian. In the group of patients>65, 101 people had already been operated on for sinus conditions, while 36 people in the control group had.
- Higher neutropihls in older patients with CRSsNP was an interesting and unexpected finding, as the authors mentioned. Please explain the reason and provide an interpretation of this finding thoroughly in the discussion section.
We are unable to clearly explain this result. Perhaps this CRS phenotype is more dependent on an ongoing inflammatory process caused by bacteria or viruses (non-Th2 phenotype).
- Please unify the acronyms of Unilateral (UNIL) or bilateral (BIL). For example, UNI and BIL.
Corrected at the manuscript.
- It seems that Table 1 is redundant because it is exactly the same as Figure 1. Please consider removing one of them. (probably table 1)
Table 1 has been removed.
- Figure 1: Please provide the unit of the Y-axis. (probably percentage)
Done.
- Figure 2 and 3: Please revise the order and the name in the X-axis. (<65 years first and ≥ 65 years later, in addition please note “≥”, not “>”)
Done.
- It is hard to find the description in the text regarding Figure 3. Please add it in detail.
It has been added
- Figure 4: It looks like there are statistical differences between CRSsNP UNIL and CRSwNP UNIL in addition to CRSsNP UNIL and CRSsNP BIL. Please clarify this.
Difference between CRSsNP UNIL and CRSwNP UNIL was not statistically significant, p = 0.09 probably due to low size of these subgroups.
- The authors set the age of 65 as the cut-off line. Please add the reference from the previous literature explaining the cut-off of old age.
The World Health Organization considers age 60 to be the beginning of old age. However, it divides this period of life into three stages: 60-75 years - old age (known as early old age), 75-90 years - old age (known as late old age), 90 years and over - old age (known as longevity). Applying the socioeconomic imperative in Europe, this is the conventional limit of 65 years (World Report on Aeging and Health.

Round 2
Reviewer 2 Report (New Reviewer)
Thank you for your reply and correction of the manuscript.
There are two same tables 3, so please delete one of them.
Author Response
Thank you for your review and reply. The table corrected at the manuscript.
Reviewer 3 Report (New Reviewer)
It is clear that the authors have made an effort to respond to each the reviewer’s suggestions, which is greatly appreciated. However, there are still several points that may require revision or clarification.
Here are some minor suggestions.
2. Although the authors explained systemic parameters in the introduction section, there is still a minor lack of information regarding the role of laboratory values, particularly in CRS. In CRS, we usually test eosinophil levels, especially when we consider qualifying patients for steroid therapy or biologic therapy. Assessment of serum IgE is important in atopic patients in whom we are considering allergen immunotherapy.
-> LINE 55: Please add the descriptions regarding the role of other systemic parameters such as WBC, ESR or CRP in CRS
LINE 215: Same table was added again without the title. Please, remove the redundant table.

Author Response
Thank you for your review and repley.
Although the authors explained systemic parameters in the introduction section, there is still a minor lack of information regarding the role of laboratory values, particularly in CRS. In CRS, we usually test eosinophil levels, especially when we consider qualifying patients for steroid therapy or biologic therapy. Assessment of serum IgE is important in atopic patients in whom we are considering allergen immunotherapy.
-> LINE 55: Please add the descriptions regarding the role of other systemic parameters such as WBC, ESR or CRP in CRS
In the study group, 5% of patients(n=20, in the >65 years old group there was 1 patient) had elevated C-reactive protein values ( norm<5mg/ml) mean 13.7+_ 10.8mg/ml (max value 52.8mg/ml). In the control group, only one patient had elevated CRP.
Elevated leukocytosis ( above 10,000/microliter) in the study group > 65 years old was found in 5 patients had values above normal (4-10 ths:) 13,090, 10,060, 11,270, 11,950, 13,490 (mean 11,972). In the <65 yr group, leukocytosis was normal, while in the control group, only 1 patient had-11,880 leukocytes/microliter.
Perhaps this was due to the eligibility conditions ( no exacerbation of chronic disease, no antibiotic therapy min 4 weeks, outside the pollen season for atopics, no foci of infection such as decayed teeth, purulent tonsils, non-healing wounds, etc.).
Absolute eosinophilia was examined in the <65 yr old group (289.7+_SD 251.8) and in the >65 yr old group (280+_SD194.5) finding no significant difference. The value for the entire study group 284 +_SD 218.7.
1 patient ( group<65 yrs)) was found to have hypereosinophilia (>1500). The value of 50-500 eosinophils in the study group was present in 82.7% of patients in the control group 87.5% . Values of 500-1500 in the study group were in 11.5% ( and in 2.8% in the control group.
There were no differences in the prevalence of atopic diseases between the studied age groups. SIT was carried out in patients < 65 years of age ( we do not have exact data, patients did not always remember the duration and type of immunotherapy because they often underwent treatment in childhood).
LINE 215: Same table was added again without the title. Please, remove the redundant table.
Corrected at the manuscript.
This manuscript is a resubmission of an earlier submission. The following is a list of the peer review reports and author responses from that submission.
Round 1
Reviewer 1 Report
Dear Editor and Authors,
Thank you for giving me the opportunity to review this paper entitled “Phenotypes of chronic rhinosinusitis and peripheral blood leukocytes parameters in elderly patients”. I have read the whole paper thoroughly with great interest. I have some remarks and I personally feel could be corrected before taking the final decision of publication.
My Specific Comments:
a) How the authors calculated/defined sample size that should be described in the method section.
b) All the figures are very poor in image quality that can be improved.
c) Reference 1 can be given in a shorter form.
d) I recommend authors to read and discuss more similar articles in the field for the improvement of background and discussion section of their article.
e) Please don’t repeat the results again and again the discussion and conclusion sections. More accurate and robust conclusion need to be made based on the present findings and discussion highlighting the practical implication of the findings and mentioning some recommendations for further improvement if similar studies in future.
Author Response
Reviewer 1:
- a) How the authors calculated/defined sample size that should be described in the method section.
It was far more common for people under 65 years of age to get CRS, so one year was enough to get a control group. Elderly patients are in the minority, so from the group of all patients admitted within 5 years (2016-2022) patients who met the accepted age criteria were selected.
- b) All the figures are very poor in image quality that can be improved.
Figures has been corrected.
- c) Reference 1 can be given in a shorter form.
References 1 has been corrected.
- d) I recommend authors to read and discuss more similar articles in the field for the improvement of background and discussion section of their article.
3 articles have been added to manuscript to improve a discussion.
- e) Please don’t repeat the results again and again the discussion and conclusion sections. More accurate and robust conclusion need to be made based on the present findings and discussion highlighting the practical implication of the findings and mentioning some recommendations for further improvement if similar studies in future.
Corrected in the manuscript in discussion and conclusions.
Reviewer 2 Report
This manuscript was entitled as “Phenotypes of chronic rhinosinusitis and peripheral blood leukocytes parameters in elderly patients”. The authors concluded that CRS in patients over 65 years of age occurs most often as a phenotype with polyps. The numbers of neutrophils and lymphocytes in the peripheral blood were significantly higher in the group ≥ 65 years of age in the phenotype of bilateral lesions without polyps.
There are several concerns in this manuscript.
1. In this study, there is lack of a healthy control group. Therefore, it is not clear whether the higher numbers of neutrophils in the peripheral blood in the group ≥ 65 years of age in CRS patients without polyps was also present in the healthy subjects with ≥ 65 years of age.
2. The authors divided their patients into 4 groups: CRS with bil. polyps, CRS with unilateral polyps, CRS without unilateral polyps, and CRS without bil. polyps. It is not clear what is the difference between CRS with unilateral polyps and CRS without unilateral polyps.
Author Response
1. In this study, there is lack of a healthy control group. Therefore, it is not clear whether the higher numbers of neutrophils in the peripheral blood in the group ≥ 65 years of age in CRS patients without polyps was also present in the healthy subjects with ≥ 65 years of age.
The authors assumed that the control consisted of younger patients with the same concomitant disease (CRS). One of the reasons for this is that it is difficult to assemble a group of healthy patients >65 years of age who have been treated in the ENT department.
2. The authors divided their patients into 4 groups: CRS with bil. polyps, CRS with unilateral polyps, CRS without unilateral polyps, and CRS without bil. polyps. It is not clear what is the difference between CRS with unilateral polyps and CRS without unilateral polyps.
The authors investigated two basis CRS phenotypes: with or without polyps. In any phenotypes, the lesions may be one-side or both – sided. CRS with one –sided polyps belongs to the plenotypes with polyps (CRSwNP), CRS without one- sided polyps CRSsNP.
Round 2
Reviewer 2 Report
This manuscript was entitled as “Phenotypes of chronic rhinosinusitis and peripheral blood leukocytes parameters in elderly patients”. The authors concluded that CRS in patients over 65 years of age occurs most often as a phenotype with polyps. The numbers of neutrophils and lymphocytes in the peripheral blood were significantly higher in the group ≥ 65 years of age in the phenotype of bilateral lesions without polyps.
1. The authors divided their patients into 4 groups: CRS with bil. polyps, CRS with unilateral polyps, CRS without unilateral polyps, and CRS without bil. polyps. CRS with unilateral polyps was supposed to be CRS with nasal polyps in one-sided nasal cavity and without nasal polyps in the other side of nasal cavity. CRS without unilateral polyps was supposed to be CRS without nasal polyps in one-sided nasal cavity and with nasal polyps in the other side of nasal cavity. I do not understand what the difference between CRS with unilateral polyps and CRS without unilateral polyps is.
Author Response
- The authors divided their patients into 4 groups: CRS with bil. polyps, CRS with unilateral polyps, CRS without unilateral polyps, and CRS without bil. polyps. CRS with unilateral polyps was supposed to be CRS with nasal polyps in one-sided nasal cavity and without nasal polyps in the other side of nasal cavity. CRS without unilateral polyps was supposed to be CRS without nasal polyps in one-sided nasal cavity and with nasal polyps in the other side of nasal cavity. I do not understand what the difference between CRS with unilateral polyps and CRS without unilateral polyps is.
The authors divided patients into two age groups and investigated the incidence of chronic sinusitis with nasal polyps (CRSwNP) or without polyps (CRSsNP) in each group.
In CRSsNP, the inflammatory lesions are either bilateral (BIL) or unilateral (UNI) only within the sinuses, e. g. isolated sinusitis, hemipansinusitis, etc. A common feature of CRS sNP bil /uni is the absence of polyps in the nasal cavity.
CRSwNP – inflammatory lesions are in the sinuses, but they are accompanied by polyps in the nasal cavity, on one or both sides.
CRSsNP UNI lesions inside the sinuses on the right or left side always without polyps in the nasal cavity, while CRSwNP UNI -inflammatory lesions in the sinuses often bilateral, but polyps always only on one side.

Round 3
Reviewer 2 Report
Phenotypically, chronic rhinosinusitis can be divided into chronic rhinosinusitis with nasal polyps or chronic rhinosinusitis without nasal polyps based on the presence of nasal polyps.